# Structure and Physical Properties of Cardamonin: A Spectroscopic and Computational Approach

**DOI:** 10.3390/molecules25184070

**Published:** 2020-09-06

**Authors:** Iwona Budziak, Marta Arczewska, Daniel M. Kamiński

**Affiliations:** 1Department of Chemistry, University of Life Sciences in Lublin, Akademicka 15, 20-950 Lublin, Poland; iwona.budziak@up.lublin.pl; 2Department of Biophysics, University of Life Sciences in Lublin, Akademicka 13, 20-950 Lublin, Poland; 3Department of Chemistry, Maria Curie-Sklodowska University in Lublin, pl. Marii Curie-Skłodowskiej 2, 20-031 Lublin, Poland

**Keywords:** chalcones, single-crystal XRD analysis, UV-Vis spectroscopy, DFT calculations, dimer

## Abstract

This is the first study of the crystal structure of cardamonin (CA) confirmed using single-crystal XRD analysis. In the crystal lattice of CA, two symmetry independent molecules are linked by hydrogen bonds within the layers and by the π···π stacking interactions in the columns which lead to the occurrence of two types of conformations among the CA molecules in the crystal structure. To better understand the stability of these arrangements in both crystals and the gaseous phase, seven different CA dimers were theoretically calculated. The molecular structures were optimized using density functional theory (DFT) at the B3LYP/6–311G+(d,p) level and the spectroscopic results were compared. It was found that the calculated configurations of dimer I and III were almost identical to the ones found in the CA crystal lattice. The calculated UV-Vis spectra for the CA monomer and dimer I were perfectly consistent with the experimental spectroscopic data. Furthermore, enhanced emissions induced by aggregated CA molecules were registered in the aqueous solution with the increase of water fractions. The obtained results will help to further understand the relation between a variety of conformations and the biological properties of CA, and the results are also promising in terms of the applicability of CA as a bioimaging probe to monitor biological processes.

## 1. Introduction

In recent decades, dietary compounds, such as naturally occurring chalcone, have enjoyed considerable interest due to the fact of their health-promoting properties. They have shown a wide range of bioactivity including antioxidant, anticancer, antimicrobial as well as anti-inflammatory properties [1,2,3]. Chalcones are assumed to serve as precursors of flavones in the biosynthesis of flavonoids. Structurally, chalcones possess a common chemical 1,3-diaryl-2-propen-1-one scaffold in which the two aryl groups are linked together by carbonyl and an α,β-unsaturated system [4,5]. In recent years, chalcones have also been applied using a technological approach such as a nonlinear optic (NLO) material [6,7]. Cardamonin (2′,4′-dihydroxy-6′-methoxychalcone, C_16_H_14_O_4_) belongs to the chemical class of chalcones which are phenolic compounds isolated from several species of *Zingiberaceae* including *Alpinia katsumadai, Alpinia conchigera, Alpinia rafflesiana, Amomum subulatum* and *Boesenbergia pandurata* [8]. The chemical structure and atom numbering are presented in Scheme 1. Cardamonin is found in cardamom spice and possesses various pharmacological properties including anticancer effects due to the cell apoptosis caused in nasopharyngeal carcinoma, prostate cancer and triple-negative breast cancer cells [9,10], inhibition of tumour incidence, tumour multiplicity, Ki-67 and β-catenin positive cells [11,12]. In addition, cardamonin exhibits antinociceptive effects in mice through the involvement of TRPV_1_, glutamate and opioid receptors [13]. It can be used in the prevention of oxidative stress-mediated neurodegenerative disorders [14].

Despite numerous studies regarding the health benefits of CA, its behaviour at the molecular level has been less thoroughly investigated. Cardamonin displays poor bioavailability that could be associated with the low water solubility relevant to its specific chemical structure and physical properties [8]. The lack of substantive spectroscopic studies concerning the physicochemical characterization of CA may cause problems with its biochemical analysis and make its application in the pharmaceutical industry difficult. Until now, only a few studies have been performed on the structural and spectral properties of natural chalcone compounds [15,16,17]. Moreover, the Cambridge Crystallographic Data Centre (CCDC) database has only included the dimethyl cardamonin (DMC) crystallographic structure of CA [18].

The present study comprised a combination of theoretical and experimental investigations of the structural and physical properties of cardamonin dimers. The molecular structures of CA dimeric species were optimized using density functional theory (DFT) at the B3LYP/6–311G+(d,p) level and compared with the single-crystal XRD analysis and spectroscopic results.

## 2. Results and Discussion

### 2.1. Crystal Structure Description

The skeleton of the CA molecule is a two-component system of aromatic rings: polyol ring (A) and phenyl ring (B) bonded by an *α*,*β*-unsaturated carbonyl system (propenon) in the *trans* configuration (Scheme 1). CA crystallizes in the monoclinic space group P21/c with cell parameters a = 25.7114(6) Å; b = 15.3506(3) Å; c = 6.8717(2) Å; β = 97.058(3)°, V = 2691.60(12) Å^3^, final R indices (I > 2σ(I)) = 0.0678 and goodness of fit (S) = 1.0594. Data relating to the final structural refinement are shown in Table 1.

The conformational variability found in two CA molecules (molecule-A and molecule-B, Figure 1) can be interpreted as a consequence of slight rotations around the C1–C10 bond axis. The values of the chosen torsion angles defined by the atoms C10–C9–C8–C7 and C9–C8–C7–C1 are 155.37° and −149.48° and 178.90° and −177.47° for molecule-A and -B, respectively. Aside from this seen in Figure 1, such conformational features can also be noticed in a superposition of both molecules found in the asymmetric unit cell. An inversion (as a mirror image) of one molecule leads to an almost prefect overlay of molecules A and B (at the bottom of Figure 2a). The angle of ~10° between the planes of the aromatic rings A and B indicates planarity of both molecules, as for the analogous structure of Flavokavain B [2]. The hydrogen bond lengths and valence angles in both CA molecules are equal (within the experimental error range) and are presented in Table 2. In the solid state, the rotation of both molecules is prevented by the O2–H2O···O1 hydrogen bond between carbonyl oxygen and hydrogen from the hydroxyl group and the π···π stacking between the CA molecules from neighbouring layers, see Figure 2c.

In the crystal lattice, molecules type A (green) and B (blue) form the stacks along the *c*-axis and interact through π–π electrons between the A-rings. The calculated distance between molecules-A in a stack is constant and equal to 3.35 Å (distance between A-rings). The stacked molecules-B are separated alternately by the distances of 3.459 Å and 3.389 Å. The molecules-A in a stack are related through glide planes, whereas molecules-B give a centrosymmetric packing. Both types of molecules interact with neighbouring stacks along the *b*-axis through intermolecular hydrogen bonds (O3–H3O···O1) extending to layers parallel to the (001) plane (see Figure 2b). The layers of pure A and pure B molecules are stacked along the *a* crystallographic axis interacting through the C–H…π type and/or other van der Waals interactions and, thus, build the 3D structure of the studied cardamonin compound.

### 2.2. Theoretical Calculations

A large number of studies concern the experimental analysis of monomeric and dimeric structures of various molecule types relative to theoretical calculations [19,20,21,22]. There have been reports on natural chalcones that form several dimeric forms and act as anti-viral or anti-cancer agents [23,24]. The spatial arrangement of aromatic rings in such dimeric structures includes hydroxyl groups which may allow them to have better interaction with amino acid residues of proteins or enzymes [25]. Therefore, to support our experimental findings, a series of calculations related to different arrangements of CA molecules (i.e., monomer and seven dimers) was performed (see Figure 3). It was found that the lowest energy was related to dimers I, II and III. These dimers corresponded to the basic blocks found in a stack of molecules-A and -B in the crystal lattice. Dimer I (head to tail) was very similar to the molecular arrangement of stacked molecules-B (blue), while dimers II and III were comparable to the conformation of CA molecules in an A-type stack shown in Figure 2c. The calculated distances between the A rings in the dimers were almost identical to those found in the crystal lattice.

For dimer I, the distance was found to be 3.35 Å (blue, Figure 2c), and in case of dimers II and III (green), the values were 3.29 Å and 3.15 Å, respectively. After the geometry optimization, dimer VI was rearranged to match the geometry of dimer II. Among all dimeric species, dimer IV had the highest energy. Due to the fact that the calculations were performed in the gas phase and the basis set superposition error differed among dimers, the calculated energy only approximately reflected the stability of the dimers (see Table 3).

The electronic absorption and excitation properties of CA were estimated by applying the time-dependent DFT approach at the B3LYP level of theory with the 6-311+G(d,p) basis set. It was found that the theoretical absorption spectra of monomer and dimer I, II and III were almost identical to those obtained from the optical absorption spectra as provided in the next section. The results of the DFT calculations of electronic transition within the range of 300 nm to 500 nm and at oscillator strengths (*f*) for CA species are shown in Appendix A.

### 2.3. Spectral Properties

The experimental UV-Vis spectrum (Figure 4a) of CA in ethanol showed the absorption maximum at 344 nm (ε = 2.8 × 10^4^ M^−1^cm^−1^) which is in excellent agreement with the computed value of 346 nm. The theoretical wavelengths were shifted towards higher values because the computations were performed in the gaseous state, whereas the observations were registered in the solution. The UV-Vis spectroscopic features are characteristic for chalcones as well as due to the strong absorption band at 340−390 nm (conjugated π system involving the B-ring) and a shoulder located at 240−300 nm (the A-ring benzoyl system, Scheme 1) [16]. The results of the spectroscopic measurements showed that in a water and ethanol (EtOH)/water mixture, the absorption band was red-shifted to the value of 374 nm and 357 nm, respectively (Figure 4b). The differences in the spectra of CA in aqueous solution and EtOH were likely caused by both solvent effects and aggregation. The hypothesis could be corroborated by the similarity between the absorption maxima of CA in aqueous solution and the crystalline state, wherein two dimeric forms were observed. These results were confirmed by theoretical calculations as provided above.

The value of direct allowed optical bandgap energy (E_g_) was determined using Tauc’s relation in a long-range absorption wavelength through extrapolation of the linear region and intercept energy axis at (αh*ν*)^2^ = 0, where h is the Planck’s constant and *ν* is the frequency of incident photons as shown in the inset of Figure 4a [26]. The experimental optical bandgap energy was found to be 3.09 eV, 2.75 eV and 2.80 eV for the monomeric form, aggregated forms in water and EtOH/water mixture (1:1), respectively. The predicted energy gaps were 3.82 eV for the monomer, 3.47 eV and 3.55 eV for dimer I and dimer III, respectively. On the other hand, the HOMO (Highest occupied molecular orbital) and LUMO (Lowest unoccupied molecular orbital) eigenvalues and their energy gaps (ΔE*_H-L_*) offered a reasonable qualitative indication of the stability and the reactivity of molecules [27]. Among the CA dimeric species, dimer I and dimer III yielded the lowest difference between HOMO and LUMO (i.e., 3.47 eV and 3.55 eV), which implies high chemical reactivity and weak kinetic stability of these forms. Consequently, a very important factor that can contribute to high reactivity is the possibility of chalcone cyclization to flavanones [18]. Our previous work related to another chalcone showed that xanthohumol exhibits short-term stability in aqueous solutions, and cyclizes to isoxanthohumol, especially under high-temperature conditions and in an alkaline medium [28]. The stability of CA during the registration of the UV-Vis spectra was monitored by checking if a new band at 330 nm, which is characteristic of the occurrence of cyclization, appeared in the spectra. It was established that the intensity of this absorption maximum increased with the progression of the cyclization process at 50 °C (Appendix A).

Moreover, it has been previously reported that the small energy gap values of chalcone derivatives contribute to their potential application as nonlinear optical materials [29]. The value of ΔE*_H-L_* and the extrapolated band gaps of the analysed CA species are presented in Appendix A.

Due to the fact of their low solubility in water, CA molecules tend to self-associate in an aqueous solution and exhibit different morphology forms, leading to altered fluorescence behaviours. In fact, as a member of the π-conjugated system with proper electron-pulling and electron-pushing functional groups on the benzene rings, chalcones have been suggested to show fluorescence properties [30]. The emission spectra of CA were studied in ethanol and an ethanol/water mixture with various water volume fraction (f_w_) to investigate this phenomenon. The fluorescence intensity gradually increased from a 0% water fraction (EtOH) up to 100% (water), then it reached its maximum (Figure 5). The emission wavelength was red-shifted from 420 to 440 nm which was attributed to the formation of aggregated structures. When the possibility for hydrogen bonding is high, chalcone molecules may combine to form different molecular forms, e.g., crystalline aggregates [31]. This process can affect the emission characteristics so that fluorescence is weakened or quenched at high local concentrations associated with the formation of aggregates. The phenomenon is widely known as the concentration quenching effect [32]. In turn, the results from our experimental study showed the opposite effect, as the emission was enhanced when the water fraction in ethanol increased. The poor solubility of CA in EtOH/water mixtures should lead to aggregation and weaken fluorescence due to the strong intermolecular π–π stacking interaction between the aromatic rings resulting in decay or non-radiative relaxation back to the ground state from the excited state [30]. Interestingly, the aggregation-induced enhanced emission from the CA molecules was not observed in the solid state (Appendix A). Indeed, in water (also in PBS buffer), the fluorescence intensity was three times higher than in ethanol. Such an aggregation-induced enhanced emission (AIEE) mechanism has already been reported for small, organic π-conjugated molecules [33,34], but its mechanism is still under debate and not completely understood. The restriction of intramolecular rotation (RIR), twisted intramolecular charge transfer (TICT) and J-aggregation are among the major factors that may be significant to explain this phenomenon [31,35,36].

## 3. Materials and Methods

### 3.1. Materials

Cardamonin with a ≥98% purity was purchased from Sigma–Aldrich (Saint Louis, MI, USA) and used with further purification by way of recrystallization from n-hexane-acetonitrile (9:1). The solution was evaporated to obtain plate-like shape crystals (0.2 × 0.1 × 0.02 mm) exhibiting a bright yellow colour. These crystals were then dissolved in ethanol and allowed to stand for 24 h at a temperature of 4 °C. Crystals were collected from the mother phase directly before measurements. All the used chemicals were purchased from Sigma–Aldrich (Saint Louis, MI, USA).

### 3.2. Methods

#### 3.2.1. UV-Vis Spectroscopy

Electronic absorption spectra of CA were recorded with a double-beam Cary 300 Bio (Varian) UV-Vis spectrophotometer. The spectra were collected at the wavelength range of 200−600 nm using a 2 nm step spacing, 1 cm long, closed quartz cuvette (Helma). The temperature (21 ± 1 °C) was controlled with a thermocouple probe (Cary Series II from Varian) placed directly into the sample.

#### 3.2.2. Steady-State Fluorescence Spectroscopy

Room-temperature fluorescence emission spectra were recorded with a Cary Eclipse spectrofluorometer (Varian, Sydney, Australia). The excitation was set at 300 nm for the solutions and at 360 nm for the solid-state measurements. The excitation and emission slits were 10 nm.

#### 3.2.3. Single Crystal X-ray Diffraction

The single crystal X-ray diffraction data for cardamonin were collected using a Rigaku XtaLAB MM7HFMR diffractometer equipped with the Cu rotated anode. Cell refinement, data collection as well as data reduction and analysis were performed with CRYSALIS Ver. 39.29d. The structure was solved by direct methods with the use of SHELXS [37] and refined using Olex2 v.1.3 program [38]. The refinement was based on squared structure factors (F^2^). All non-H atoms were refined with anisotropic displacement parameters. Almost all hydrogen atoms were located in the idealized geometric positions except those forming hydrogen bonds. Table 1 includes experimental details for all measured single crystals. The crystallographic data were deposited at the Cambridge Crystallographic Data Centre (CCDC) under No. 2014912.

#### 3.2.4. Gaussian16 Calculations

All calculations were performed in the Gaussian 16. The structures of CA dimers were optimized with the use of the DFT/B3LYP method with the 6-31+(p,d) atomic basis set. In all calculations, empirical dispersion with the Beck–Johnson damping scheme GD3BJ was introduced to describe weak interactions between molecules in a dimer. Tight convergence criteria were used in the calculations during geometry optimization. The excited states were treated with the TDDFT framework [39]. Vertical electron transitions from the ground state to the excited state were calculated using the linear response (LR) method.

## 4. Conclusions

In general, chalcones show poor solubility in water solutions. This could be explained by the dimerization process which can further lead to aggregation. We found that cardamonin in water solution exists in a dimeric state, as opposed to EtOH, where CA molecules exist mainly as monomers. The calculated spectra for the CA monomer and dimer I are in excellent agreement with the experimental data obtained from crystals and solutions. Moreover, the structures calculated for dimers I and III are a very similar to the conformations found in the CA crystal lattice. Interestingly, CA emits only weak fluorescence in pure ethanol, whereas a tendency for enhanced emissions was observed for self-associated CA molecules in solutions with increasing water fractions. Based on the results, it can be concluded that the CA molecule should be suitable for potential bioimaging applications in living systems.

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
