# Peer review of "Structure and Physical Properties of Cardamonin: A Spectroscopic and Computational Approach"

_molecules, 2020, doi:10.3390/molecules25184070_

Round 1

Reviewer 1 Report

The authors present the  crystal structure study of the cardamonin compound, crystallized from a commercially available product. Also they give results concerning the energetically favored overlap of neighboring molecules, and they complete their manuscript with UV-Vis spectroscopic and fluorescent studies. The work worth to be published but with major revision.

Following are my comments that might help the authors to improve their manuscript:

1st comment: In line 18 (Abstract) they say that C-H..π interactions involving the centroids of phenyl rings stabilize the crystal net. Nowhere in the text they even mention this type of interactions. Also, the use of word dimer in the title or elsewhere in the text is confusing. Is better not to mention this feature of cardamomin molecule as this characteristic concerns different conformation of it (have a look at the 4th of my comments bellow).

2nd comment: It will become clear later of this review (in a next of my comments), that in the asymmetric unit exist two symmetry independent molecules with different conformations. This point has to be stated clearly at this point of the abstract.

  3rd comment: According to the list of tips for finalizing refinement (https://journals.iucr.org /services/cif/tips.html) , the labeling scheme that have been followed by the authors contain superfluous characters. Please number the atoms as C1, C2, C3.., O1, O2..etc and follow a consistent scheme for both molecules.

4th comment: For the section 2.1, Crystallographic structure, is better to use the title Crystal structure description. The authors do not discuss at all the different conformations they get from their structure analysis, for the two symmetry independent molecules. To my opinion the best way to describe this difference is by using the torsion angle defined by the atoms C00M, C00J, C00I, C009 for the first molecule (molecule-A) and C00T, C00V, C00O, C00P for the second one (molecule-B). In the first case this angle takes the value 178.90 and in the second -177.5. An overlapped diagram of the two molecules would be very helpful in presenting both conformations. As the only known analogous compound is the dimethyl one [ref 15 in the present work] it would be interesting if the authors discuss also the conformations observed in the structure of the dimethyl compound.  This is an interesting point of the present study because this feature of the molecule i.e. to present a variety of conformations, make it susceptible to polymorphism.

5th comment:  The part of section 2.1, lines 84-90, where the authors give a description for the packing of the structure is confusing. They must specify which one of the two molecules presented in Figure-1 are the green and the blue ones in Figure-2.  The blue molecules in Figure-2 correspond to those which contain the C00T carbon atoms (the molecule at the right side of Figure-1, molecule-B) and the green one corresponds to molecule which contains C00M carbon atoms (the molecule at the left side of Figure-1, molecule-A). To my opinion   the packing of the molecules have to be described as follows: Molecules-A are stacked along C-axis and neighboring molecules along the stack are centrosymetrically related and interact though π-π interactions and molecules-A at neighboring stacks along b-axis interact through the hydrogens bonds O003–H003···O005, ….. Thus molecules of type A form layers parallel to the (100) crystallographic plane. Also molecules of type B form  stacks along c-axis  through π-π interacting  molecules which are related through mirror plane symmetry and neighboring stacks interact through O008–H008···O007, …. hydrogen bonds extending to layers parallel to the (100) planes. The layers of pure A and pure B molecules are stacked along the a crystallographic axis interacting through C-H…π type and or other type of van-der Waal interactions and thus build the 3D structure of the studied cardamomin compound.  The different conformations of the two molecules give a different way of packing. Is the interplanar spacing along each stuck constant or dimmers are formed, which the usual case?

6th comment: Do the authors have taken in to account, in their theoretical calculations the different conformations and the symmetry relations of neighboring molecules.

7th comment: The authors have to make a comment for the A-Alert they get in the check cif file and also to add the color and the dimensions of the studied crystal in the cif file.

8th comment: The physical properties are well described and well presented.

Author Response

Response to Reviewer #1

First of all, we would like to express our deepest gratitude to the Reviewer for the valuable guidance which undoubtedly has enriched this manuscript. We agree with most of the comments. All proposed suggestions have been included in the new version of the manuscript.

Major points:

Point 1: In line 18 (Abstract) they say that C-H..π interactions involving the centroids of phenyl rings stabilize the crystal net. Nowhere in the text they even mention this type of interaction. Also, the use of word dimer in the title or elsewhere in the text is confusing. Is better not to mention this feature of cardamomin molecule as this characteristic concerns different conformation of it (have a look at the 4th of my comments below).

Response 1: Thank you very much for your valuable comments. Firstly, regarding the sentence “…C-H..π interactions involving the centroids of phenyl rings stabilize the crystal net”, we agree with the Reviewer and a confusing sentence was removed from the text. Moreover, we decided to provide an additional explanation in the abstract (lines 16-24).

According to the next part of the Reviewer`s comments, the necessary changes with using “dimer” in the title and in the text, but only regarding cardamonin structure description have been made.

Point 2:  It will become clear later of this review (in the next of my comments), that in the asymmetric unit exist two symmetry independent molecules with different conformations. This point has to be stated clearly at this point in the abstract.

Response 2: The correction has been made. The sentence now reads: “In the crystal lattice of CA, two symmetry independent molecules are linked by hydrogen bonds within the layers and by the π···π stacking interactions in the columns, which leads to the occurrence of two types of conformations between the CA molecules dimers in the crystal structure.”.

Point 3: According to the list of tips for finalizing refinement (https://journals.iucr.org /services/cif/tips.html) , the labeling scheme that have been followed by the authors contain superfluous characters. Please number the atoms as C1, C2, C3.., O1, O2..etc and follow a consistent scheme for both molecules.

Response 3: Firstly, we are really sorry that we did a mistake with the labeling scheme. In the revised manuscript, atom numbering has been changed according to your suggestion in Figure 1, and in parts associated with new labeling, also in a CIF file.

Point 4: For the section 2.1, Crystallographic structure, is better to use the title Crystal structure description. The authors do not discuss at all the different conformations they get from their structure analysis, for the two symmetry independent molecules. To my opinion the best way to describe this difference is by using the torsion angle defined by the atoms C00M, C00J, C00I, C009 for the first molecule (molecule-A) and C00T, C00V, C00O, C00P for the second one (molecule-B). In the first case this angle takes the value 178.90 and in the second -177.5. An overlapped diagram of the two molecules would be very helpful in presenting both conformations. As the only known analogous compound is the dimethyl one [ref 15 in the present work] it would be interesting if the authors discuss also the conformations observed in the structure of the dimethyl compound.  This is an interesting point of the present study because this feature of the molecule i.e. to present a variety of conformations, make it susceptible to polymorphism.

Response 4: As suggested by the Reviewer, we have made a correction in the title of section 2.1. Additionally, we added a discussion about the different conformations of CA molecule (lines 97-106) and an overlapped diagram in Figure 2a. In this work, we would like to focus on the effect of dimerization in solution and two molecular arrangements, therefore, we did not discuss the conformations observed in the structure of the dimethyl CA compound.

Point 5_1:  The part of section 2.1, lines 84-90, where the authors give a description for the packing of the structure is confusing. They must specify which one of the two molecules presented in Figure-1 is the green and the blue ones in Figure-2.

Response 5_1: You are right. It was an unfortunate oversight. This part (lines 111-120) of text was completely changed according to the suggestion.

Point 5_2:  The different conformations of the two molecules give a different way of packing. Is the interplanar spacing along each stuck constant or dimmers are formed, which the usual case? 

Response 5_2: The calculated distance between molecules-A in a stack is constant and equal to 3.35Å (distance between A-rings). The stacked molecules-B are separated alternately by the distances of 3.459 Å and 3.389 Å. We discussed a different way of packing in lines 114-116.

Point 6:  Do the authors have taken in to account, in their theoretical calculations the different conformations and the symmetry relations of neighboring molecules.

Response 6: The theoretical calculations consider only the dimer structures in the gaseous phase.

Point 7:  The authors have to make a comment for the A-Alert they get in the check cif file and also to add the color and the dimensions of the studied crystal in the cif file.

Response 7: We are really sorry, we did a mistake with the preparation of cif file, but that's a terrible oversight on our part. The correction has been made and a Corrected cif file was redeposited in CCDC database under the same number.

Point 8:  The physical properties are well described and well presented.

Response 8: Thank you for the valuable comments, we appreciate your time and the effort you have made to review our paper.

Reviewer 2 Report

This paper reports the study of the crystal structure of cardamonin confirmed using single crystal XRD analysis. Hydrogen bonds and π···π stacking interactions lead to the occurrence of two types of dimers in the crystal lattice of CA.

The molecular structures of CA dimeric species were optimized by theoretical calculations related to seven different arrangements of CA molecules, using density functional theory (DFT) at the B3LYP/6–311G+(d,p) level, and compared with the single-crystal XRD analysis and spectroscopic results.  

Altough  I don't feel qualified to judge about the English language and style, I think that the paper must be thoroughly revised. The introduction is too verbose and could be siutably reconsidered and made mor easily readable.

As an example, all the description from line 52 to line 66 seems largely not interesting.

In conclusion, I think that the paper can be accepted for publication after minor revisions.

Author Response

Response to Reviewer #2

Thank you for the valuable comments, we appreciate your time and the effort you have made to review our paper. The article was corrected according to the comments provided in the review and all the statements were addressed below.

Point 1. Although  I don't feel qualified to judge about the English language and style, I think that the paper must be thoroughly revised.

Response 1: The new version of the manuscript has been revised according to your suggestion.

Point 2. The introduction is too verbose and could be suitably reconsidered and made more easily readable. As an example, all the description from line 52 to line 66 seems largely not interesting.

Response 2: The introduction has been corrected and the necessary changes have been made that now it is more acceptable and readable (lines 56-64). More to the point, we decided to remove a part of a confusing description.

Round 2

Reviewer 1 Report

The authors have addressed all the points made by the reviewer   and I think that the manuscript in its current form have substantially improved. I suggest the publication of the manuscript in its present form.